# High-resolution ultramicroscopy of the developing and adult nervous system in optically cleared *Drosophila melanogaster*

Marko Pende[1,2], Klaus Becker [1,2], Martina Wanis[1,3], Saiedeh Saghafi[1], Rashmit Kaur[3], Christian Hahn[1,2], Nika Pende[4], Massih Foroughipour[2], Thomas Hummel[3] & Hans-Ulrich Dodt [1,2]

The fruit fly, *Drosophila melanogaster*, is an important experimental model to address central questions in neuroscience at an organismic level. However, imaging of neural circuits in intact fruit flies is limited due to structural properties of the cuticle. Here we present a novel approach combining tissue clearing, ultramicroscopy, and data analysis that enables the visualisation of neuronal networks with single-cell resolution from the larval stage up to the adult *Drosophila*. FlyClear, the signal preserving clearing technique we developed, stabilises tissue integrity and fluorescence signal intensity for over a month and efficiently removes the overall pigmentation. An aspheric ultramicroscope set-up utilising an improved light-sheet generator allows us to visualise long-range connections of peripheral sensory and central neurons in the visual and olfactory system. High-resolution 3D reconstructions with isotropic resolution from entire GFP-expressing flies are obtained by applying image fusion from orthogonal directions. This methodological integration of novel chemical, optical, and computational techniques allows a major advance in the analysis of global neural circuit organisation.

[1] Department for Bioelectronics, FKE, Vienna University of Technology, Gußhausstraße 25-25A, Building CH, 1040 Vienna, Austria. [2] Section of Bioelectronics, Center for Brain Research, Medical University of Vienna, Spitalgasse 4, 1090 Vienna, Austria. [3] Department of Neurobiology, University of Vienna, Althanstrasse 14, 1090 Vienna, Austria. [4] Department of Ecogenomics and Systems Biology, Archaeal Biology and Ecogenomics Division, University of Vienna, Althanstrasse 14, 1090 Vienna, Austria. These authors contributed equally: Marko Pende, Klaus Becker. Correspondence and requests for materials should be addressed to M.P. (email: marko.pende@tuwien.ac.at) or to H.-U.D. (email: hans.ulrich.dodt@tuwien.ac.at)

In recent years, the fast development of tissue-clearing approaches has offered a wide range of tools for studying deep-tissue structures[1–5,6–11,12–14], large neuronal projections, entire organs, and even whole animals[15–20]. The main focus of these approaches was to visualise tissue organisation in vertebrate model organisms and humans. If animals genetically modified for tissue-specific fluorescence transgene expression were not available, the samples had to be immunohistochemically labelled, which is a difficult, laborious, and time consuming task for large specimens[21–24].

Regarding the fast generation time, large numbers of externally laid embryos, and the capability of being genetically modified in numerous ways, the fruit fly Drosophila melanogaster has proved to be a valuable model organism to address fundamental questions in a variety of biological processes[25]. In multiple disciplines of Drosophila research, the most commonly used imaging technique is confocal microscopy following tissue dissection[26]. However, when large undissected volumes are recorded, the drawbacks of confocal microscopy are the low imaging depth, the long recording time, and the occurrence of photo bleaching. Without clearing of the sample, strong absorption and scattering of photons occur by interaction with the tissue, especially with the pigments located in the eyes and the cuticle. These aberrations result in a decrease of signal intensity and non-uniform spatial resolution[27,28]. Tissue transparency is usually achieved by reducing the refractive index (RI) gradient at the borders between different cellular components.

Commercially available mounting media, such as DPX, Glycerol, 2,2′-Thiodiethanol (TDE)[29], VECTASHIELD®, FocusClear®, ProLong™Gold, RapidClear®, Histodenz™ or recently published direct immersion media such as FRUIT[6], ClearT[7], ScaleS[8], SeeDB[9], SeeDB2[10] and RTF[11] have such RI matching properties. However, each of these solutions has some limitations regarding the preservation of morphology, fluorescent stability, viscosity, penetration depth and the capacity to render tissue transparent[8,10,11]. Further, none of these media has been reported to have depigmentation properties. Therefore, heavily pigmented organisms, such as Drosophila require sample preparation including the dissection of organs before mounting and imaging, which leads to tissue damage and deformation. For example, sensory neurons located in peripheral structures like the eye, antenna, or legs, establish long-range projections via various nerves to the central brain, where they segregate into distinct synaptic regions. Following the dissection and mounting of peripheral and central brain tissues the location of sensory neurons and their brain connections cannot be visualised in total so far. Here, critical information about overall neural map organisation, especially the spatial relationship between cell body position and synaptic contacts, is lost by animal dissection. Furthermore, digital reconstruction of separate images is known to be quite labour-intensive and error-prone.

The importance of visualising distinct cell types in their organismal context has led to various attempts to image tissue organisation during development in intact Drosophila in the past[30–32]. Despite some progress in the field, previous approaches in whole fly imaging have shown some critical drawbacks, including the limitations of specific cell labelling[33–35], the loss of fluorescent signal, the necessity of immunohistological labelling, low spatial resolution and the need for specialised equipment, e.g., the head-array preparation[30]. As previous studies have shown that the autofluorescence of optically cleared flies can be imaged with micrometre-resolution in a short period of time by light-sheet microscopy[36], an efficient tissue-clearing technique, which preserves the stability of endogenously expressed fluorescent proteins would be a key innovation to overcome current technological disadvantages.

Here, we present a novel clearing protocol, FlyClear, which results in optical transparency of intact Drosophila while preserving the fluorescence of endogenously expressed green fluorescent protein (GFP) and mCherry.

To gain higher spatial resolution during ultramicroscope imaging, we used an improved light-sheet generator that produces a thinner light-sheet with a much-extended length of uniformity along the propagation axis compared to a standard light-sheet generator system composed of a slit aperture and a cylindrical lens[36].

In order to achieve a virtual isotropic resolution of the recorded data in all spatial directions, we implemented a multi-view combining algorithm computing a three-dimensional (3D) Fast-Fourier Transformation (FFT) to identify the sharper regions in a set of image stacks recorded from different spatial directions. Although combining multi-view microscopy image stacks has been done before[37–43], to the best of our knowledge, it was never used in optically cleared specimens.

By combining our innovations in clearing technique, light-sheet imaging, and computational multi-view combining, we show for the first time the systemic 3D imaging of endogenous fluorescent markers in intact D. melanogaster for multiple developmental stages and adult flies with isotropic spatial resolution and long-term signal stability.

## Results

**Optical tissue clearing with FlyClear**. D. melanogaster is a challenging subject for optical tissue clearing using previously established protocols[9,12–44]. The structural features of the chitinous exoskeleton and various kinds of pigmentation, including the photopigments within the compound eye, impede the clearing process as well as the RI matching to obtain optical transparency. Furthermore, expressed fluorescent proteins in transgenic fruit flies are more sensitive to chemical treatment than in other model organisms such as mice[32,36].

To overcome these drawbacks, we developed a clearing method called FlyClear. Starting with the clear, unobstructed brain imaging cocktails and computational analysis (CUBIC) protocol[14], we applied additional tissue treatment steps, depending on the developmental stage of the flies, and several modifications to the original reagent-1. For larva and prepupa we used a 0.03% Protease treatment step prior to formaldehyde fixation. In addition, a permeabilisation step with acetone was applied for prepupa after formaldehyde fixation (Fig. 1a). The concentrations of the detergent and the used aminoalcohol in the new solution were significantly decreased (hereafter termed as Solution-1). A substantial change in Solution-1 (see Methods) from the original CUBIC reagent-1 was the replacement of Quadrol® (N,N,N′,N′-tetrakis(2-hydroxypropyl)-ethylenediamine) by THEED (2,2′,2″,2‴-(Ethylenedinitrilo)-tetraethanol), which was also tested in the original CUBIC paper. This adjustment not only accelerates the clearing process, but also results in a complete depigmentation of the compound eye (Supplementary Fig. 1a) and a sufficient decrease in the cuticle pigmentation (Fig. 1b).

To test, whether the signal quenching effect caused by THEED, as described before[14], is too strong for transgenic fly lines with a low fluorescence expression level we directly compared the signal stability in uncleared and Solution-1 cleared half's of same flies with weak GFP expression in the antenna (50 neurons).

Compared to the control half's we observed a significant loss of GFP signal. However, due to the increased transparency and reduced autofluorescence we had no difficulties to image the neurons with the same settings as the uncleared ones (Supplementary Fig. 1b, c).

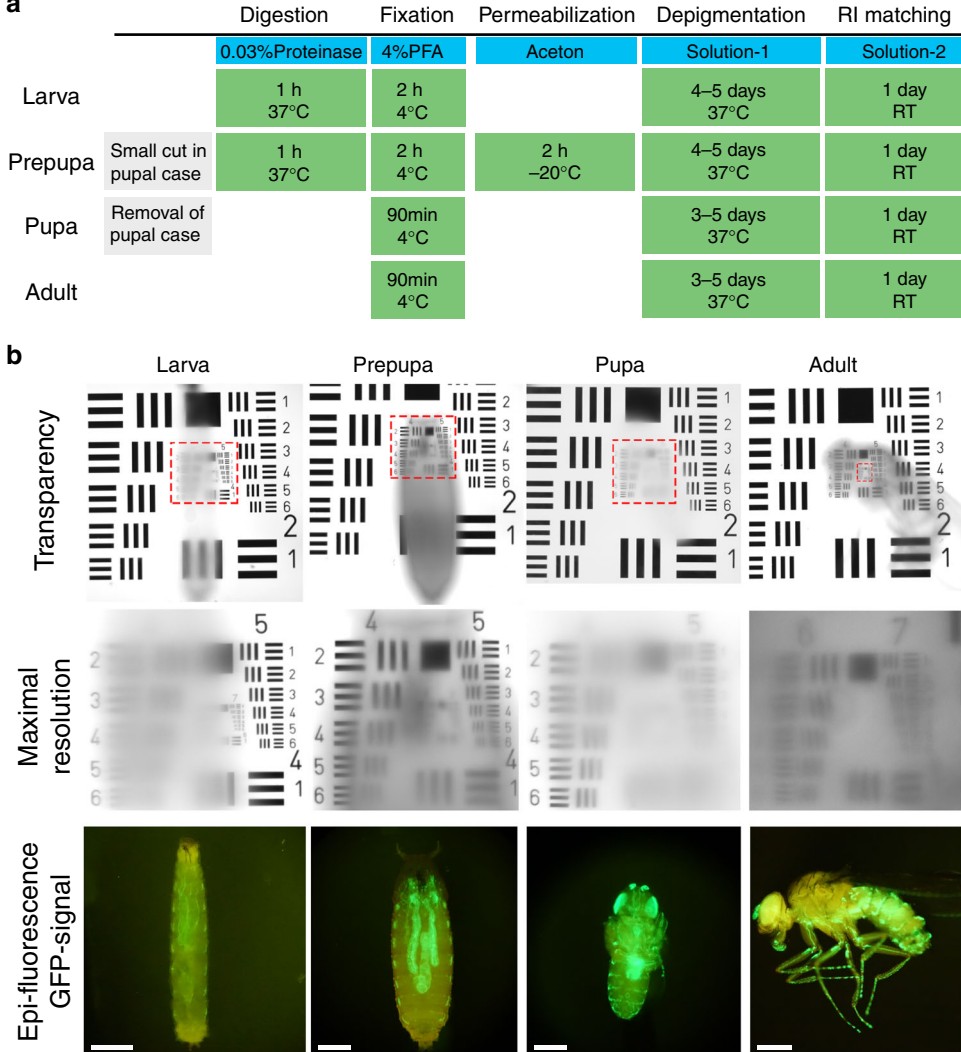

**Fig. 1** Workflow and properties of the FlyClear procedure. **a** Main steps of the FlyClear protocol. **b** Upper panel shows wide-field image of optically cleared specimens placed on top of a USAF1951-chart in Solution-2 demonstrates the level of overall transparency. Middle panel shows higher magnification of red rectangular areas indicating the highest level of transparency reached in the corresponding sample. Lower panel shows GFP signal after RI matching. Bright field images were acquired with a 4x objective (Olympus, XLFluor4x/340, 0.28 NA, WD = 29.5 mm) using custom-made correction of optics for a refractive index of 1.45 (WD = 10 mm after correction). Fluorescent images were acquired with a stereomicroscope with a 1x objective (Leica, Plan APO 1.0×, WD 61.5 mm). Genotype: *Peb-Gal4 UAS-mCD8::GFP;*. Scale bars represent 500 μm in **b**

The fluorophore quenching effect of Solution-1 led us to test alternative RI matching media (Fig. 1b and Supplementary Figs. 1d, e and 2). Ultramicroscope image quality is dependent on the uniformity and viscosity of the imaging medium, e.g., streaks in the liquid can have distortion effects on the light-sheet. Furthermore, viscous liquids can cause tissue deformation due to high osmolarity difference between the tissue and its surrounding. To prevent such effects we decided to use an aqueous solution of meglumine diatrizoate, a component which is also used in FocusClear™[45]. Among the tested RI media, this solution gave us the best results regarding, morphology preservation, sample transparency and liquid viscosity (Supplementary Fig. 1e and 2). Meglumine diatrizoate is an aromatic iodine compound that can be easily dissolved in water or phosphate buffered saline (PBS) to create a solution with a final RI of 1.45 (hereafter termed Solution-2). Additionally, it has to be mentioned that no volume change could be observed when comparing cleared with the corresponding uncleared side of the flies (Supplementary Fig. 1d, e).

We tested the signal stability of GFP following incubation in Solution-2 (see Methods) by recording samples at three different time points under the same conditions (Supplementary Figs. 3 and 4a). We did not observe any significant changes in fluorescence intensity of individual flies even after 30 days (Supplementary Fig. 4b, c). Further, we tested the protocol on mCherry labelled samples and demonstrated the signal preservation of an alternative fluorophore (Supplementary Fig. 5a, b).

With the resulting final protocol, we were able to generate significantly improved transparency of larval and pupal specimens and highly transparent samples of adult flies (USAF1951-chart: group 7, element 6 = 2.19 μm resolution) (Fig. 1b and Supplementary Table 1) with stable fluorescence intensities over several weeks (Supplementary Fig 4).

**New aspheric ultramicroscope system with improved light-sheet.** A thinner light-sheet gives a better resolution in *z*-direction. Therefore, we tried to decrease the full width half maximum

(FWHM) of our light-sheet to increase the isotropy of imaging in $x$, $y$ and $z$, allowing to view the sample from different directions with similar resolution.

We used ultramicroscopy for virtual sectioning to achieve high-resolution images from our cleared *Drosophila* samples (Supplementary Fig. 6a, b). A cylindrical lens in combination with a rectangular slit aperture is the most common approach for generating a thin light-sheet in ultramicroscopy. Therefore, we refer to this as the standard ultramicroscope set-up. The process of the optimisation of the ultramicroscope involves the usage of various optical components (Fig. 2a). These components change the phase and other related factors of the incident beam to reshape it to an ultra-thin beam with improved characteristics. Combinations of two or three cylindrical lenses can affect the orbital angular momentum of the laser beam and enables us to alter the distribution of the output beam[46]. However, some limitations and side effects may arise if aberration and related parameters are not controlled during this process[36]. One of the most significant reasons for using Aspheric elements in any optical design is their ability to form a distortion-free image with minimal aberration. A laser beam with a symmetrical Gaussian intensity profile can be converted into a semi-uniform beam by using two identical Aspheric lenses with a particular surface design[36] (Figs. 2a (1) and (3)). Placing a Meso-Aspheric element, such as a Powell lens (Fig. 2a (2)), that generates a beam with semi-uniform intensity distribution between two Aspheric lenses facing each other enabled us to produce an elliptical beam with a uniform intensity profile. A thin light-sheet can be produced if this beam is incident on a unit containing two identical Acylinder lenses of focal length $f$ that are separated by $\sqrt{2}f$ (Fig. 2a (5) and (6))[36].

The output beam can be further improved by using a custom-made elliptical apodising soft-aperture (Fig 2a (4)). A mathematical function representing an elliptical Bullseye filter was obtained by using the size and shape of the elliptical uniform beam when the beam radius was at its minimum value before the beam changed the direction of its distribution toward the converting unit containing two Acylindrical lenses. Thus, a custom-made soft-aperture was designed and allocated at a distance of $2F_c$, with $F_c$ being the focal length of the third optical element. The soft-aperture enables us to eliminate the unwanted intensity distribution in our optical system without creating the hard-edge aperture effects that we encounter in the conventional truncation. This alteration is primarily dependent on the transmission gradient of the aperture. The combination of these complex optical elements redistributes the amplitude and phase of the beam in a way that generates a light-sheet with a highly localised intensity at the focal point and a vastly extended length of uniformity along all axes. This is achieved while simultaneously minimising the power loss when compared with the standard ultramicroscope set-up.

Using LaserCam-HRTM (Coherent, Germany), the intensity profiles across the $x$-$y$ plane produced by the standard ultramicroscope set-up and the aspheric ultramicroscope at three different positions (focus, 1000 μm, and 2000 μm away from the focus) were measured (Fig. 2b, c). Based on these measured data, the FWHM values were determined (Fig. 2b, c).

**Imaging in intact Drosophila larval and adult flies**. We reasoned that the characterisation of complex organisms require methods for imaging multi-tissue interactions of distant connections in intact samples. We used FlyClear in combination with the improved ultramicroscope set-up and confocal microscopy to visualise structures of the sensory-, secretory-, respiratory- and reproductive system of intact *Drosophila* expressing *UAS-mCD8::*

*GFP* under the developmental driver line Pebbled-Gal4[47]. In 3rd instar larvae internal tissues like trachea, mid gut and salivary glands can be visualised (Supplementary Fig. 7a, b and Supplementary Movie 1). In prepupa, the projection of photoreceptors in the developing visual system (Fig. 3a, b) and the innervation of the segmental nerves into the ventral nerve cord become visible (Fig. 3b–e and Supplementary Movie 2). In mid pupal stages, we demonstrated the high spatial resolution by imaging neurons of the visual-, reproductive- and the olfactory system (Supplementary Fig. 8a–e). Additionally, we visualised the neurons in various appendages of pupae and adult flies (Supplementary Fig. 8f and 9a, b and Supplementary Movie 3).

**Precise neuronal circuit mapping in developing and adult flies**. A major advantage of FlyClear is the complete removal of eye pigments in *Drosophila* pupal and adult stage, thereby allowing a detailed analysis of individual neuronal populations and their connectivity without brain dissection. To demonstrate the spatial resolution at different levels of the central brain we imaged two neuron types in the *Drosophila* visual system, the dorsal cluster neurons (DCNs) and medulla columnar neurons (MCNs). In wild type, a bilateral cluster of DCNs in the dorso-lateral central brain forms an interhemispheric commissural tract to innervate the medulla and lobula synaptic neuropils in the fly optic lobes (Fig. 4a, c)[48]. In homozygous mutants of the neuronal cell adhesion molecule Neuroglian, the specific loss of the commissural tract can be detected following the FlyClear treatment of intact *Drosophila* (Fig. 4b, d and Supplementary Movie 4). Similarly, for a homogenous group of MCNs in the adult visual system a single synaptic layer in the central medulla can be separated from the neuronal cell bodies in the peripheral medulla cortex (Fig. 4e, f) further illustrating the high spatial resolution in the analysis of neural circuit organisation in the intact fly.

In addition, we showed with our aspheric ultramicroscope system the organisation of the undissected visual system of pupa (Supplementary Fig. 8a, b and Supplementary Movie 5).

Until now, the majority of protocols for investigating neuronal connections in *D. melanogaster* involve brain dissection[49]. This leads to disrupted connections of peripheral neurons into the central brain, impeding a complete understanding of sensory map formation. Therefore, high-resolution phenotypic analysis of wild-type and mutant neural circuit organisation is impaired. Applying the FlyClear protocol in combination with improved ultramicroscopy, we are able to obtain a complete visualisation of neuronal structures leading from the cell bodies in the periphery to the synaptic target regions in the central brain (Fig. 5).

For the first time, it was possible to visualise the undissected projection of sensory neurons from their receptors in eyes, antennae, maxillary palps and labellum (Fig. 5a) to their respective central processing areas in the brain (Fig. 5b and Supplementary Movie 6).

We could clearly resolve small neuronal connections (Fig. 5a, b and Supplementary Fig. 10). We were also able to follow the projections of the antennal nerve up to the antennal lobe in detail (Fig. 5c, d and Supplementrary Movie 7). Furthermore, we demonstrated that the protocol enables counting of individual cell bodies in the intact *Drosophila* (Figs. 4c, d, e, 5a, b inset and Supplementary Fig. 8bf and 10 inset).

**Multi-view combining for 3D-reconstruction**. Generally, the axial resolution is lower than the lateral resolution in both confocal and light-sheet microscopy[37,39,40]. To achieve 3D-reconstructions with virtual isotropic resolution, we recorded flies from two orthogonal directions and combined the stacks using a multi-view combining approach[37,39,40] (Supplementary

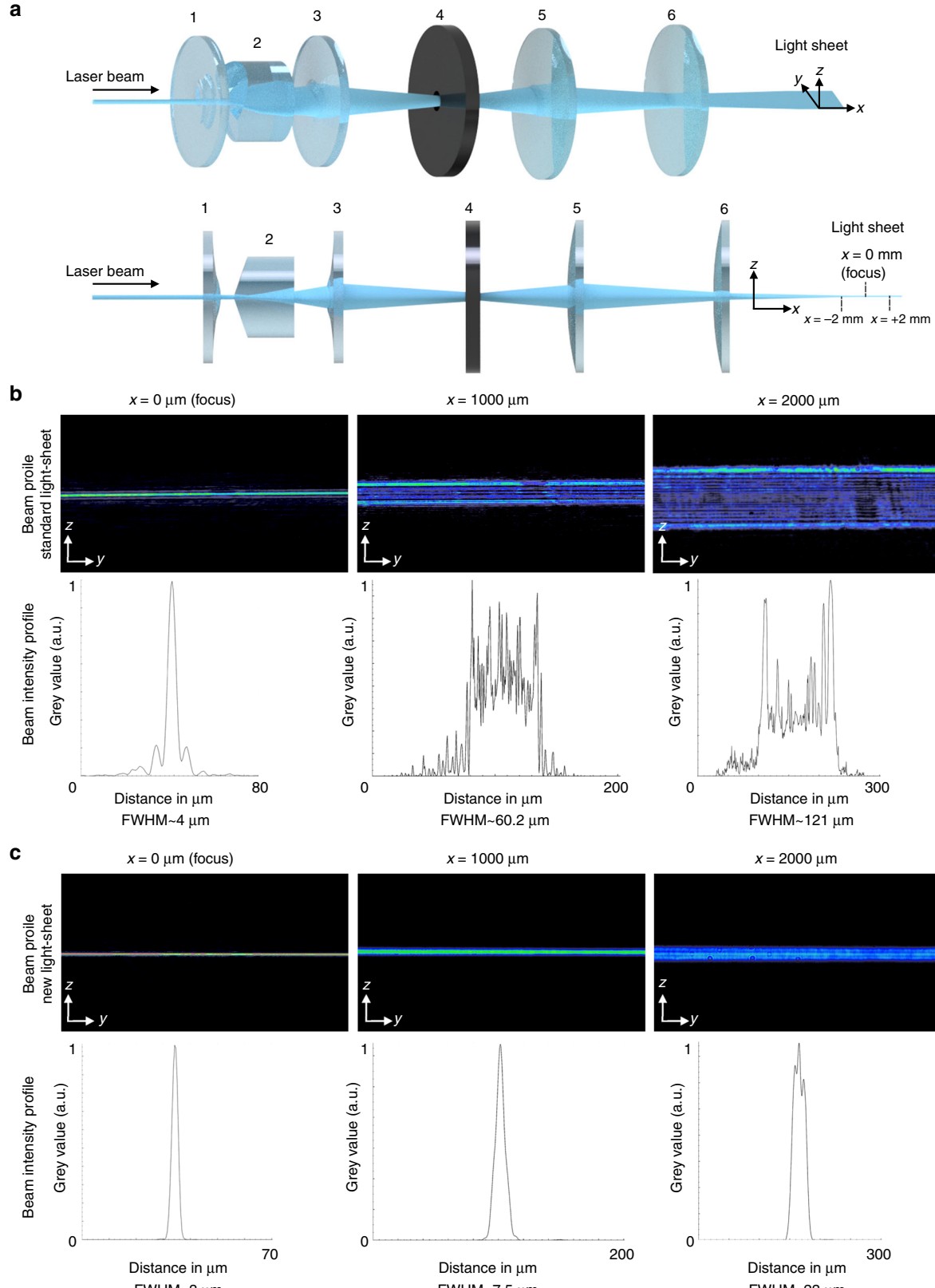

**Fig. 2** Characterisation of the improved aspheric ultramicroscope system. **a** Illustration of the shaping of the laser beam to a light-sheet by the optimised optical unit containing complex optical components. Plano-convex aspheric cylinder lenses (1, 3), Powell lens (2), an elliptical soft-aperture (4) and Acylinder lenses (5, 6). **b**, **c** Difference between the intensity distributions across the z-y plane measured at the position x on the light-sheet by LaserCam-HRTM. Standard ultramicroscope with a 8 mm-wide slit aperture (**b**) and the aspheric ultramicroscope system optimised with a soft-aperture (**c**) measured at the focus x = 0, x = 1000 μm and x = 2000 μm

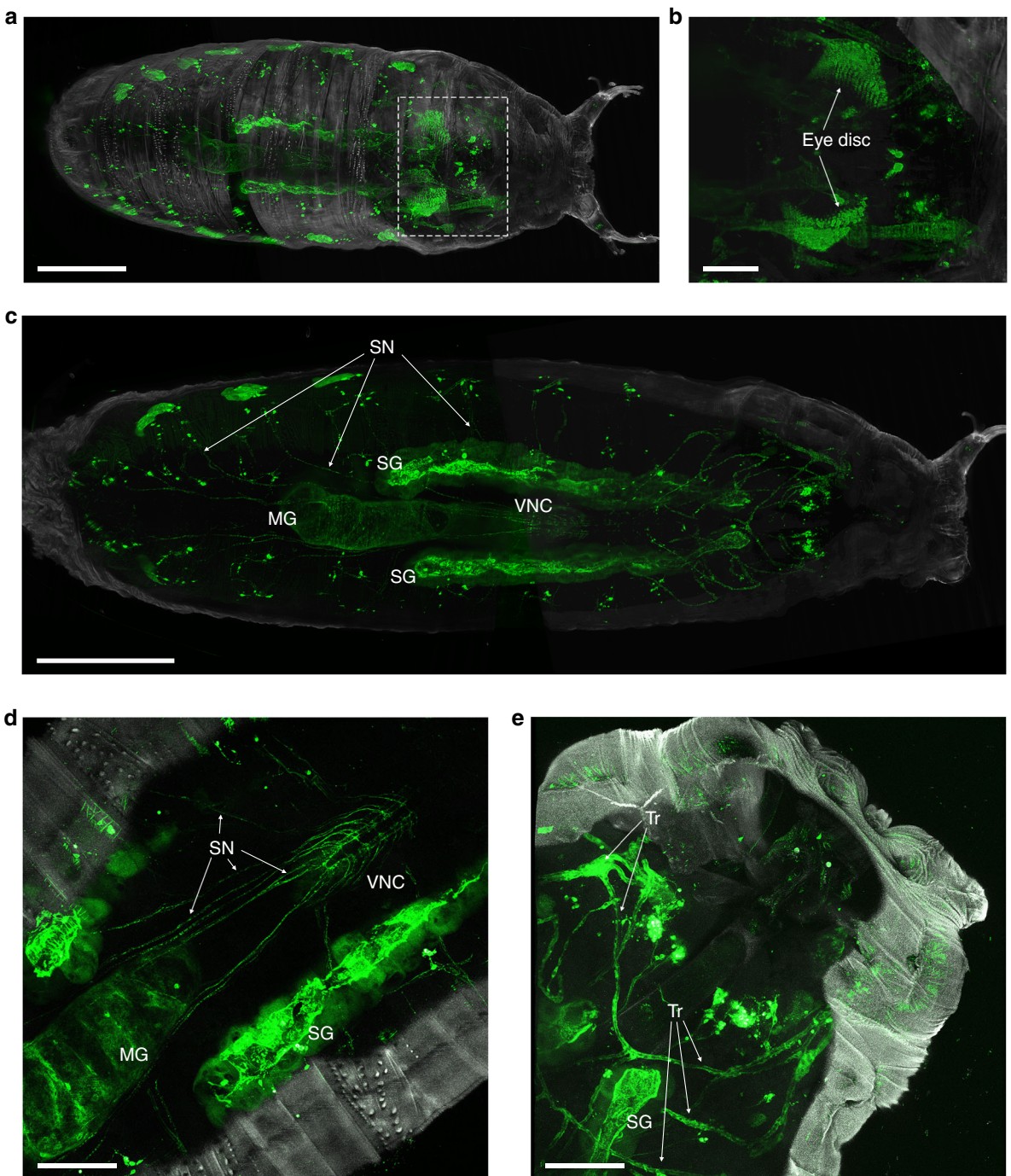

**Fig. 3** Imaging of FlyClear treated prepupa using confocal and optimised aspheric ultramicroscope system. **a–c** Light-sheet images and **d**, **e** confocal images of GFP expression in the developing visual system, segmental nerves (SN), and trachea (Tr). **a** Imaging of whole prepupa. **b** Clipping plane shows higher magnification of boxed area in **a** displaying the eye discs of prepupa. **c** Dorsal view of a clipping plane in undisected prepupa showing mid gut (MG), salivary glands (SG), segmental nerves (SN) and the ventral nerve cord (VNC). **d** Higher magnification showing the innervation of the SN into the VNC. **e** Higher magnification showing trachea (Tr) and salivary glands (SG). Images in **a–c** were acquired with a 10x water-immersion objective (Olympus, UMPlanFLN, 0.3 NA, WD = 3.5 mm) with custom-made correction of optics for a refractive index of 1.45 (WD = 3.5 mm after correction). Images **d**, **e** were acquired with a 20x immersion objective (Leica, HCX PL APO CS, 0.7 NA, 260 µm WD). Genotype: *Peb-Gal4 UAS-mCD8::GFP*. Scale bars represent 500 µm in **a**, **c**, 200 µm in **c**, **d** and **e**

Fig. 11). The algorithm for this method uses a 3D FFT to identify parts that are sharper in one of the two pre-aligned image stacks. Once these parts are identified, it combines them to a fused image stack. As already suggested by Shaw et al. and Sätzler and Eils[37], frequency components with higher magnitudes are assumed to represent the sharper image structures. Therefore, the multi-view combined stack was obtained by an inverse FFT after selecting for the frequency components with these higher magnitudes and

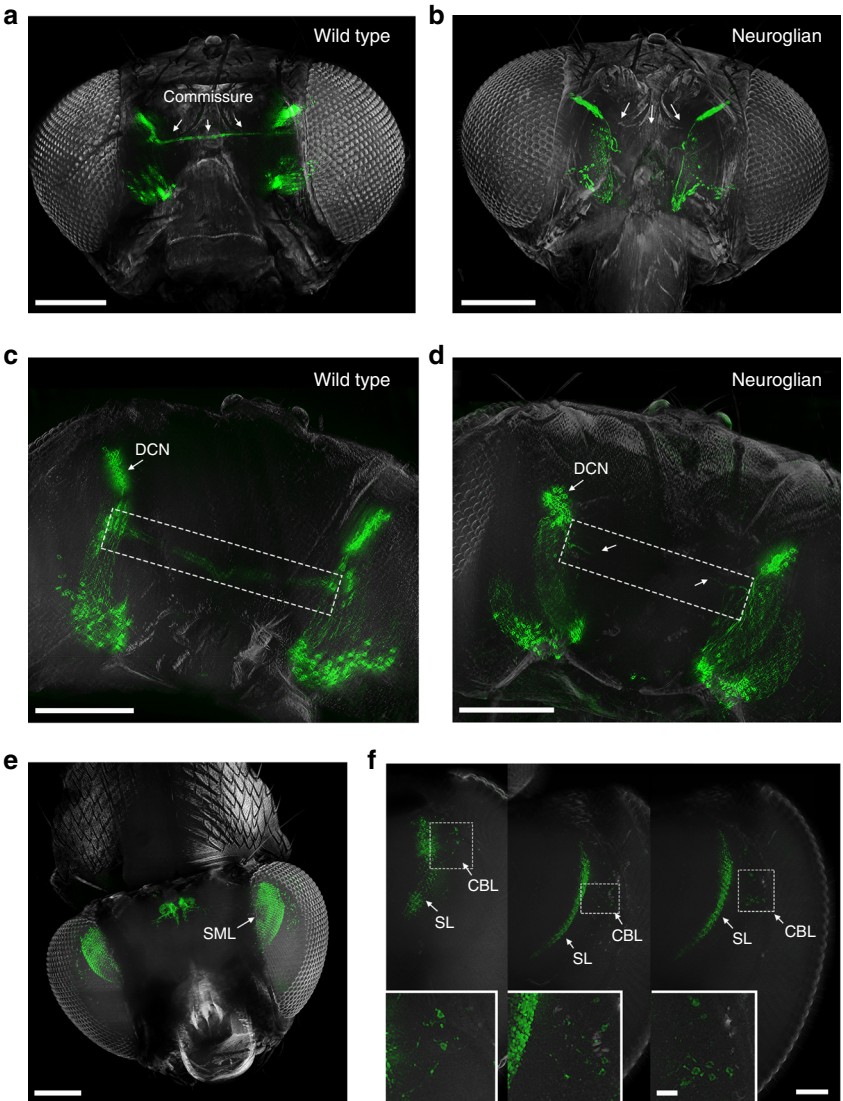

**Fig. 4** Imaging of the *Drosophila* visual system of a FlyClear treated intact adult fly using optimised aspheric ultramicroscope system. **a–f** Light-sheet images of GFP expression in visual system. **a–d** Analysis of interhemispheric circuit defects in Neuroglian mutants. **a**, **c** In wild type, clusters of commissural neuron (arrowhead indicate Dorsal cluster neurons, DCNs) in the visual system project across the midline in the central brain (dashed box). Loss of Neuroglian does not affect the ipsilateral organisation of visual neurons but deletes the commissural projections (arrowheads in **b**), with a few single fibres sending short processes towards the midline (arrowheads in **d**). **e**, **f** Synaptic layer organisation in the visual system. Columnar interneurons innervating a single medulla layer (SML) are visualised (**e**). Different optical sections indicate separated synaptic (SL) and cell body layers (CBL) (**f**). Images in **a**, **b** and **e** were acquired with a 10x water-immersion objective (Olympus, UMPlanFLN, 0.3NA, WD = 3.5 mm) with custom-made correction of optics for a refractive index of 1.45 (WD = 3.5 mm after correction). Images in **c**, **d** and **f** were acquired with a 25x objective (Olympus, XLPlanN, 1.0 NA, WD = 8 mm). Genotypes: **a**, **c** *Control–nrg849/ + ; UAS-mCD8::GFP; ato-Gal4 (Females)*, **b**, **d** *Mutant-nrg849/y; UAS-mCD8::GFP; ato-Gal4 (Males)* and **e**, **f**; *10x UAS-mCD8:: GFP; R47G08-Gal4*. Scale bars represent 200 μm in **a**, **b** and **e**, 100 μm in **c**, **d**, 50 μm in **f** and 10 μm in insets in **f**

adjusting the corresponding phase values (Fig. 6 and Supplementary Fig. 11). The better resolution in 3D can be best appreciated in the movie in the Supplementary (Supplementary Movie 8).

## Discussion
Here, we present a novel experimental approach that allows the visualisation of the distinct peripheral and central components of neuronal networks in intact, optically cleared *Drosophila* flies in high spatial resolution.

The FlyClear protocol provides several advantages: (1) it is a fast and simple procedure that only requires brief immersion of the flies in two solutions. If confocal microscopy is used,

Solution-2 can be replaced by VECTASHIELD®. (2) It completely maintains the morphology of the fly tissue along with high transparency and concomitant depigmentation. The strong pigmentation of the compound eye is a major obstacle as it causes absorption and diffraction of the light. FlyClear overcomes this drawback without the use of bleaching reagents that quench endogenously expressed fluorescent proteins. (3) Using this clearing protocol, the endogenous fluorescence signal is preserved for at least a month without significant changes in intensity. Some dehydration-based immersion clearing approaches for *Drosophila* have previously been published[32,36,50]. However, it was not proven whether they are able to stabilise transgenic fluorescent markers such as GFP. The images presented in these publications show neither detailed structures nor GFP signal clearly

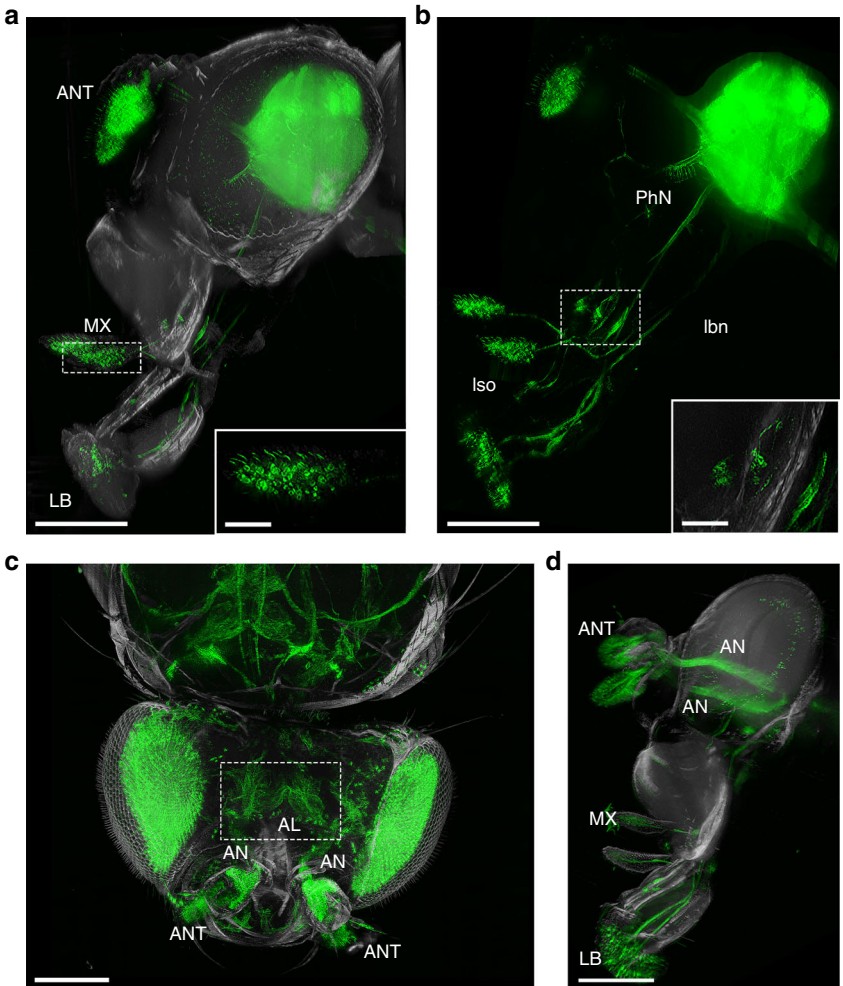

**Fig. 5** Imaging of the *Drosophila* chemosensory system of a FlyClear treated intact adult fly using optimised aspheric ultramicroscope. **a–d** Light-sheet images of GFP expression in sensory neurons. **a** Lateral view of olfactory and gustatory receptor neurons in antenna (ANT), maxillary palp (MX) and labellum (LB). The inset shows a higher magnification of the maxillary palp. **b** Pharynx and the connection of the pharyngeal nerve (PhN) with the ventral cibarial sense organ (vcso), the labral sense organ (lso), and the connection of the labellum with the labial nerve (lbn). The inset shows a higher magnification of the vcso with single neurons. **c** Dorsal view of antennal receptor neurons, which are connected with the antennal lobe (AL) (dashed box) through the antennal nerve (AN). **d** Lateral view of the connection described in **c**. Images in **a–c** were acquired with a 10x water-immersion objective (Olympus, UMPlanFLN, 0.3 NA, WD = 3.5 mm) with custom-made correction of optics for a refractive index of 1.45 (WD = 3.5 mm after correction). In **a**, **b** a post-magnification of 2x was used. Image in **d** was acquired with a 0.5x post-demagnification in combination with a 25x objective (Olympus, XLPlanN, 1.0 NA, WD = 8 mm). Genotype: **a**, **b** *dscam-Gal4/CyO; UAS-mCD8::GFP* and (**c–e**) *Peb-Gal4 UAS-mCD8::GFP*. Scale bars represent 200 μm in **a–d** and 50 μm in the magnified areas of **a**, **b**

distinguished from autofluorescence[32]. Therefore, our protocol represents a major breakthrough in the field of optical tissue clearing for insects.

We utilised an aspheric ultramicroscope system, which was improved with a soft-aperture. With this optimised system we could exhibit a light-sheet with a very low thickness at the focus with a strongly extended length of uniformity along the propagation axis and an almost uniform intensity distribution along the imaging plane.

Unlike other fluorescence based imaging techniques such as knife-edge scanning-, confocal- or two-photon excitation microscopy, ultramicroscopy of optically cleared samples allows the visualisation of intact neuronal networks without mechanical disruptions. Neurites can be imaged and traced along their projections from their receptor to their interconnections in the central brain with single-cell resolution. As a proof of principle, we visualised axonal projections of various sensory receptor

neurons and central brain populations in the adult *Drosophila* head and associated appendages to their synaptic regions in the central nervous system. Furthermore, due to the limited exposure to laser light, a sample can be repetitively imaged over a long period of time. X-ray-based methods such as MicroCT X200 or synchrotron X-ray tomography could provide a powerful alternative for whole system imaging. However, at the current state such approaches still have limitations regarding the labelling of specific cell types, tissue morphology preservation and the accessibility to a broad group of scientists.

We achieved an isotropic resolution in the 3D-reconstruction by recording cleared samples from orthogonally tilted directions and subsequently combining these tilted views into a single image stack by an FFT-based multi-view fusion algorithm. Due to the uniform resolution, it is possible to quantify cell bodies of the receptors. Our approach suggests that image fusion can be an efficient tool in general for creating isotropic

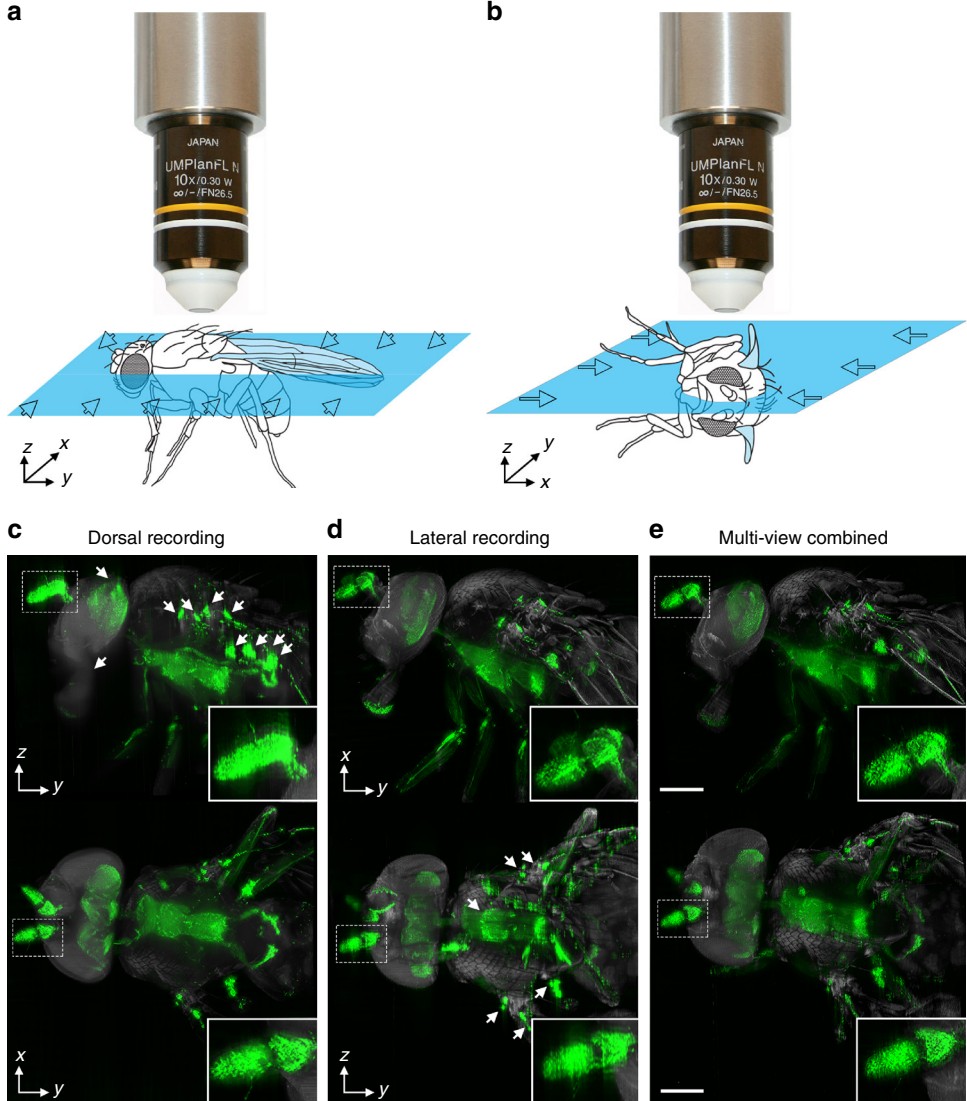

**Fig. 6** Multi-view combining of ultramicroscope recordings from orthogonal directions. **a**, **b** Illustration of the two sided light-sheet illumination through the specimen. Illumination direction is indicated by blue arrows **c**, **d** Lateral (upper panel) and dorsal (lower panel) view of ultramicroscope recordings from FlyClear processed, GFP-expressing samples. Unsharp areas are indicated by arrowheads. **a** Scheme of the lateral illumination plan and **c** corresponding recording of *D. melanogaster*. Insets show bigger magnification of antenna. **b** Scheme of the 90° tilted illumination plane and **d** corresponding recording of the specimen. Insets show higher magnification of the antenna. **e** Multi-view combined reconstructions of the stacks in **c** and **d**. Insets show higher magnification of the antenna. GFP signal is represented in green and autofluorescence is represented in grey. All images were acquired with 2x post-magnification in combination with a 4x Objective (Olympus, XLFluor4x/340, 0.28NA, WD = 29.5 mm) with custom-made correction of optics for a refractive index of 1.45 (WD after correction 10 mm). Scale bars represent 200 µm

resolution in ultramicroscope recordings from optically cleared samples.

We anticipate that the powerful combination of optical tissue clearing, light-sheet imaging, and multi-view combining will provide major benefits in different applications. It will fill the void of whole system approaches in *D. melanogaster* by enabling isotropic imaging from single neuronal projections to the entire nervous system. Therefore, the described protocol could potentially provide answers to questions addressing neuronal map formation. Additionally, other cell populations within and outside of the nervous system could be visualised in intact flies. This could enable better approaches to study various biological questions such as neurodegenerative diseases. Complete new mechanistic insights should also be possible within the field of developmental biology.

## Methods

**Animal**. All fly stocks were maintained in standard fly food and kept on 25 °C. Adult flies of 4–5 days were used in this study. Six different Gal4 drivers were analysed:

> -Peb-Gal4 UAS-mCD8::GFP;;
> -; dscam-Gal4/CyO; UAS-mCD8::GFP
> -nrg849/ + ; UAS-mCD8::GFP; ato-Gal4,
> -nrg849/y; UAS-mCD8::GFP; ato-Gal4,
> -; 10x UAS-mCD8::GFP; R47G08-Gal4,
> -; UAS-mCD8:GFP; Or47b-Gal4
> -;; R88E12-Gal4 UAS-mCherry.

These Gal4 drivers were kindly provided by Thomas Hummel, Department of Neurobiology, University of Vienna, Austria.

**FlyClear solution preparation**. Solution-1 was prepared by mixing 8 wt% THEED (2,2′,2″,2‴- (Ethylenedinitrilo)-tetraethanol) (Sigma-Aldrich, 87600–100 ML), 5wt % Triton® X 100 (Roth, 3051.2) and 25 wt% Urea (Roth, X999.2).

Solution-2 consists of 50 wt% Meglumine diatrizoate (Sigma-Aldrich M5266) in PBS (pH 8.5) adjusted to a RI of 1.45.

**FlyClear protocol for larva.** Samples were treated at 37 °C for 1 h with 0.03% proteinase (Sigma, P8038–250MG) in prewarmed PBS to digest the larva superficially. For optical clearing, larva were fixed in 50 ml of 4% PFA (pH 8.5) at 4 °C for 2 h under gentle shaking followed by 3x washing with PBS at 4 °C for 20 min each. The samples were then immersed in 10 ml of Solution-1 at 37 °C under gentle shaking for 4–5 days. The animals were then washed 3x with PBS for one day at 25 °C. To avoid deformation of the sample, the larva were incubated for 3 h in a diluted Solution-2 (25 wt% Meglumine diatrizoate). Finally, the samples were immersed in Solution-2 for 24 h at 25 °C and kept in Solution-2 at 25 °C for storage.

**FlyClear protocol for prepupal stage.** A small cut was made between T1 and T2 of each prepupal case to enable better penetration of chemicals and to avoid tissue deformation, which can be caused by the later clearing procedure. Samples were treated at 37 °C for 1 h with 0.03% proteinase (Sigma, P8038–250MG) in pre-warmed PBS to digest the pupal case superficially. Prepupa were then fixed in 50 ml of 4% PFA (pH 8.5) at 4 °C for 2 h under gentle shaking followed by 3x washing with PBS at 4 °C for 20 min each. To permeabilise the prepupal case, the samples were treated for 2 h at −20 °C in aceton before immersion in 10 ml of Solution-1 at 37 °C under gentle shaking for 4–5 days. The animals were then washed 3x with PBS for one day at 25 °C. Finally, the samples were immersed in Solution-2 for 24 h at 25 °C and kept in Solution-2 at 25 °C for storage.

**FlyClear protocol for pupa.** First the pupal case was removed while the pupa was immersed in PBS. For optical clearing, pupa were fixed in 50 ml of 4% PFA (pH 8.5) at 4 °C for 90 min under gentle shaking followed by 3x washing with PBS at 4 °C for 20 min each. The samples were then immersed in 10 ml of Solution-1 at 37 °C under gentle shaking for 3–5 days. The animals were subsequently washed 3x with PBS for one day at 25 °C. Finally, the samples were immersed in Solution-2 for 24 h at 25 °C and kept in Solution-2 at 25 °C for storage.

**FlyClear protocol for adult flies.** For optical clearing, the adult flies were fixed in 50 ml of 4% PFA (pH 8.5) at 4 °C for 90 min under gentle shaking followed by 3x washing with PBS at 4 °C for 20 min each. The samples were then immersed in 10 ml of Solution-1 at 37 °C under gentle shaking for 3–5 days, depending on the depigmentation level of the compound eyes. The animals were then washed 3x with PBS for one day at 25 °C. Finally, the samples were immersed in Solution-2 for 24 h at 25 °C and kept in Solution-2 at 25 °C for storage.

**Ultramicroscopy.** Cleared samples were illuminated from two sides using one of two Sapphire lasers, 488 nm/200 mW and 532 nm/200 mW (Coherent, Germany), producing a beam with Gaussian intensity distribution. A 50% beam splitter is used to divide the incident beam into two identical beams. They are guided towards two light-sheet generator units. The details of the light-sheet generators units and the custom-made soft-aperture are (1) Aspheric lens ($f = 20$ mm, Linus, Germany), (2) Powell lens (10° Fan-angle, Edmund optics, Germany), (3) Aspheric lens ($f = 20$ mm, Linus, Germany), (4) Bullseye filter with elliptical aperture (Reynard Cooperation, USA), (5) Acylindrical lens ($f = 80$ mm, Linus, Germany), (6) Acylindrical lens ($f = 80$ mm, Linus, Germany). The description of the light-sheet generator is given in the patent DE 102010046133B4 and in Saghafi et al.[36].

The light-sheet generators are placed on a computer-controlled linear stage (LS-65, PI-Micos, Germany), which can be moved along the beam propagation axis generating a light-sheet with an optimal line of focus propagating through the sample.

A computer-controlled elevation stage with an adjustable precision less than 100 nm (Es-100, PI-Micos GmbH, Germany), and a manually adjustable $xy$-cross table for horizontal adjustment is used to scan the sample vertically.

The light detection part includes a customised microscope with modified objectives for RI mismatch, a computer-controlled filter wheel with different optical band pass filters for blocking the fluorescence excitation light, and a scientific grade sCMOS camera (Neo, Andor, Ireland).

A custom-made software allows us to perform an automatised recording of stacks of images.

Objectives used:

-4x objective (Olympus, XLFluor4x/340, 0.28 NA, WD = 29.5 mm) using custom-made correction of optics for a RI of 1.45 (WD = 10 mm after correction)
-10x water-immersion objective (Olympus, UMPlanFLN, 0.3 NA, WD = 3.5 mm) with custom-made correction of optics for a RI of 1.45 (WD = 3.5 mm after correction)
-25x objective (Olympus, XLPlanN, 1.0 NA, WD = 8 mm) with adjustable RI

**Fluorescence stereomicroscopy.** Cleared and uncleared *D. melanogaster* were immersed in Solution-2 and imaged with Leica MZ 16F fluorescence

stereomicroscope using a one fold magnifying, long distance objective (Planapo 1×, 0.28 NA, WD = 55 mm).

**Laser-scanning confocal microscopy.** Samples treated with Solution-1 and washed with PBS were mounted in VECTASHIELD® antifading mounting medium (Vector laboratories, H-1200). *D. melanogaster* were imaged with an inverted laser-scanning confocal microscopy system (Leica, SP5) using a 20x immersion objective (Leica, HCX PL APO CS, 0.7 NA, 260 µm WD), 40x Oil-immersion objective (Leica, HCX OL APO CS, 1.25 NA, 100 µm WD) and a 63x glycerol objective (Leica, HCX PL APO CS, 1.3 NA, WD 280 µm).

**Image processing.** Image processing and 3D-reconstruction was done using Amira (FEI, USA) and Photoshop (Adobe, USA) running on a Dell workstation equipped with two 8 core Xeon processors with 256GB RAM and a Nvidia Quadro M6000 graphics card.

The images were post processed by blind maximum likelihood deconvolution, adaptive histogram equalisation, and final unsharp masking. Stripes originating from the image recording procedure were removed using a Fast-Fourier-Transform-based spatial filtering approach.

**Multi-view combining.** To achieve a virtual isotropic resolution from all directions, we applied a multi-view combining approach. For this purpose, a fly was recorded from different orthogonal directions resulting in pairs of image stacks, $I_1$ and $I_2$, which were tilted by approximately 90° (Supplementary Fig. 11). Using Amira 5.3, the two corresponding stacks were visually pre-aligned and then precisely registered with the volume registration tool. The image stack $I_2$ was re-sampled to obtain virtual slices that are coplanar with respect to the recording planes from stack $I_1$. After registration, the non-overlapping parts were removed from both stacks (Supplementary Fig. 11). To this purpose, a binary mask was generated from Stack $I_1$ by thresholding the intensity values. Then stack $I_2$ was multiplied with this mask. A second mask was obtained in the same way from stack $I_2$ and multiplied with stack $I_1$ (Supplementary Fig. 11).

To correct possible brightness differences, we normalised the processed data sets to their 95% intensity percentile before further processing. We then applied a 3D FFT to obtain the two 3 D arrays, $M_1$ and $M_2$, and the two 3D arrays that comprise the phase values, $\varphi_1$ and $\varphi_1$, from the transformed data sets $F_1$ and $F_2$:

$$M\left(F_{x,y,z}\right) = \sqrt{\Re^2\left(F_{x,y,z}\right) + \Im^2\left(F_{x,y,z}\right)}, \qquad (1)$$

and

$$\varphi\left(F_{x,y,z}\right) = \arctan\left(\frac{\Im\left(F_{x,y,z}\right)}{\Re\left(F_{x,y,z}\right)}\right) \qquad (2)$$

($\Re$: real parts of the FFT data, $\Im$: imaginary parts of the FFT data)

Assuming that the highest magnitude value at corresponding positions $x$, $y$, $z$ within $M_1$ and $M_2$ represents the sharper structural information[37,38], the magnitudes $M_{comb}$ of the multi-view combined image were calculated as:

$$M_{comb} = \max\left(M_{I_1}, M_{I_2}\right). \qquad (3)$$

The corresponding phases were obtained by adding the complex numbers from $F_1$ and $F_2$:

$$\varphi_{comb} = \arctan\left(\frac{\Im(I_1) + \Im(I_2)}{\Re(I_1) + \Re(I_2)}\right). \qquad (4)$$

The FFT $F_{comb}$ of the multi-view combined stack is obtained from the recombined magnitudes $M_{comb}$ (Eq. 3) and the recombined phases $\varphi_{comb}$ (Eq. 4):

$$\Re_{comb} = Mag_{comb} \cdot \cos\left(\varphi_{comb}\right), \qquad (5)$$

and

$$\Im_{comb} = Mag_{comb} \cdot \sin\left(\varphi_{comb}\right). \qquad (6)$$

The results from Eqs. (5) and (6) were transformed back into the spatial domain, rescaled, and saved back as 16-bit tiff-files. The images were reloaded into Amira and used for generating the final multi-view reconstructed image stack.

**Quantification of transparency.** We placed the sample in Solution-2 on a USAF1951-chart. A grayscale picture was recorded with a 4x objective (Olympus, XLFluor4x/340, 0.28 NA, WD = 29.5 mm) using custom-made correction of optics for a RI of 1.45 (WD = 10 mm after correction). The transparency was determined by examination of an USAF1951-chart through the optically cleared fly according

to Supplementary Table 1. This allowed the quantification of the resolution that was possible in the specimen.

**Quantification of fluorescence signals in Solution-1**. We fixed adult flies, with a weak GFP expression in the antenna (about 50 neurons), in 4% PFA pH 8.5. We then embedded them in 2% of low melting agarose and cut them with a vibratome in two almost identical half's. One of each half was incubated for four days in Solution-1 the other was stored in PBS pH 8.5 on 4 °C. Because light-sheet imaging of large samples needs a certain degree of transparency, we used a confocal microscope to quantify and compare the fluorescent signal loss. We mounted the untreated and Solution-1 treated samples ($n = 3$) on one slide in VECTASHIELD®, because of its fluorescence stabilising properties, and imaged them with the same settings. It has to be mentioned, that we adjusted the laser intensity always on the untreated samples to avoid overexposure. The recorded image stacks were 3D reconstructed using Amira software. For signal quantification, we calculated brightness intensity histograms within regions of interest, each comprising areas in the antenna of uncleared and solution-1 treated half's of the same fly (Supplementary Fig. 1b). We plotted the averaged intensities of the image stacks recorded at the two different conditions and calculated the standards deviations. Significance was assessed with the unpaired t-test.

**Quantification of fluorescence signals in Solution-2**. We mounted Dscam-Gal4/CyO; UAS-mCD8::GFP flies on a needle tip with UV-glue (Bondic®, Canada) and immersed them in a cuvette filled with Solution-2 (Supplementary Figs. 3 and 4). To obtain standard conditions, we recorded stacks of 550 images from each entire fly using constant laser power utilising an ultramicroscope -system equipped with an improved light-sheet generator and a 4x objective (Olympus, XLFluor4x/340, 0.28 NA, WD = 29.5 mm) using custom-made correction of optics for a RI of 1.45 (WD = 10 mm after correction). We repeated the recordings after one week and one month using identical illumination and camera settings. The recorded image stacks were 3D reconstructed and spatially registered using Amira software. For signal quantification, we calculated brightness intensity histograms within regions of interest, each comprising identical areas of the same fly recorded at three subsequent time points. (Supplementary Fig. 4).

We calculated the average intensity of all image stacks recorded at subsequent days from each fly. To represent identical exposure conditions, these results were averaged for each group of samples and plotted against the day of exposure (Supplementary Fig. 4c).

**Statistical analysis**. In Fig. 1c, data are presented as average ± s.d. For significance, *P*-values were calculated using a one-way ANOVA with IBM SPSS statistics software.

**Code availability**. The source code to perform the multi-view combining in this study is available in the Supplementary Software.

## Data availability
The data supporting the findings of this study are available from the corresponding authors upon reasonable request.

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

## Acknowledgements

We thank all lab members at the Centre for Brain Research, Medical University Vienna, the Department of Bioelectronics, Technical University Vienna and the Department of Neurobiology, University of Vienna, in particular T. Zrzavy and S. Hametner for the help with the statistics, M. Bradl and H. Lassmann for providing the incubators, A. Tröscher and J. Bauer for the help with the confocal microscope, G. Goyal, L. Timaeus, L. Geid for discussion and B. Woller, B. Camurdanoglu, S. Papadopoulos and L. Hogdskiss for proofreading. The study was funded by the Austrian Science Fund (FWF), Project P 23102-N22 and Project P 25134.

## Author contributions

H.U.D., T.H. and M.P. designed the study. M.P. developed the FlyClear protocol and performed most of the experiments. K.B. implemented the multi-view combining. M. W. developed the *Drosophila* fixation protocol. S.S. optimised the ultramicroscope and corrected the objectives for a refractive index of 1.45. R.K. provided the *Drosophila* lines. C.H. did the movies. N.P. made the *Drosophila* illustrations and M.F. made the light-sheet generator illustrations. H.U.D., T.H., M.P., K.B., M.W., N.P., C.H. and S.S. wrote the manuscript. All authors discussed the results and commented on the manuscript text.

## Additional information

**Competing interests:** The authors H.U.D., S.S. and K.B. hold a patent on the light-sheet generator (DE 102010046133B4). The other authors declare no competing interests.

