## [Peer Review File · Nature Communications]

Reviewers' Comments:

Reviewer #1:

Remarks to the Author:

This contains a nice set of technology developments to visualize fluorescently labeled signals in the intact whole *Drosophila*, which is an important experimental model. Although each technological development is rather incremental, combination of all these improvements presents a powerful and synergistic package to break technical barriers in the *Drosophila* whole 3D imaging. For example, this new technique can be quite useful to study neuronal connectivity in the intact organism and quantify its change in various experimental conditions. Virtual isotropic resolution from orthogonal imaging is also quite nice. However, I have few concerns over the technique.

Major comments.

- There is no mention or measurement of 3D volume changes after the clearing. Some clearing techniques induce some significant and non-uniform volume changes in different parts of the body. It would be important to measure this potential distortion from the clearing method.

- Although preservation of fluorescent signal after solution 2 appears to be good, there is no quantitative measurement of fluorescent signal loss before and after the clearing method. For example, the author can dissect the fly into half, measure fluorescent signal in one side before clearing and the other side (anatomically matching) after clearing. This will help to understand quantitatively how much fluorescent signal loss is expected from the clearing procedure and to determine whether weakly expressed fluorescent signals will be still preserved.

Minor comments

- At the end of the UM development description result section for Figure 2, it would be useful to include a sentence or two to explain what is the benefit of having lower FWHM in the new configuration for the imaging quality.

Miscellaneous

- typo in page 6 line 24, should be Fig 1b
- page 13, line 4, typo
- Figure 1B bottom row, no scale bar

Reviewer #2:

Remarks to the Author:

Drosophila is an important model organism in biology. While whole-brain fluorescence imaging and circuit mapping is possible with existing mounting media and confocal microscopy, it has been challenging to image *Drosophila* at a whole-body scale. For example, it has been difficult to examine the neuronal connectivity between the periphery and CNS.

To image the entirety of the *Drosophila*, the major obstacle was the chitinous exoskeleton and various kinds of pigmentation present in cells, particularly at the adult stage. In this study, the authors used a modified CUBIC method to establish an optimized clearing agent for *Drosophila* (named FlyClear). FlyClear efficiently removed pigments allowing for whole-body clearing. The authors also tried to improve the imaging resolution in ultramicroscopy from two different approaches. Firstly, authors introduced aspheric elements to make a thinner light sheet in ultramicroscopy. To further improve the z resolution, the authors also introduced a multi-view combining approach.

While each of the three components is a relatively small improvement over existing techniques, the

combination is providing the powerful pipeline for the high throughput whole-body imaging of *Drosophila*. This should become an important contribution to the field of neuroscience and developmental biology using *Drosophila*.

I suggest following minor points to improve the readability, and for the broader usage of this technique in the community.

- 1) It would be helpful if authors could include an image of the adult *Drosophila* before and after the clearing with FlyClear.
- 2) Authors should mention that the THEED used in FlyClear Solution 1 was originally described in the CUBIC paper.
- 3) Figure 2 needs a better explanation for a general readership. It may be unclear to most readers whether z is for the detection lens or illumination lens. The schema might better include the detection system with axis information, in case some of the readers are not familiar with the ultramicroscopy. It is also important to mention the objective lenses and their imaging area (x-y) for the fluorescence detection, because this information directly relates to the z value in this figure.
- 4) Please provide the refractive index of FlyClear (solution 2 for mounting), as this is important information to correct for spherical aberrations.
- 5) It is not always apparent in the figures whether the image was taken with ultramicroscopy or confocal. The microscopy type should be labelled within each figure (with objective lens information). To improve the transparency of the paper, please include a table summarizing the imaging conditions (objective lens, ultramicroscopy/confocal, etc).
- 6) Related to 3), as many of the researchers in the field only have experiences with confocal microscopy, it would be helpful if authors could provide a side-by-side comparison (resolution, imaging speed, etc.) of confocal vs ultramicroscopy in the supplement.
- 7) In the introduction, authors should mention common mounting/clearing media that were previously used (e.g., Vectashield, FocusClear, SeeDB2, DPX) and their limitations.
- 8) Be consistent throughout the text on the use of technical terms, particularly for "ultramicroscopy" and "Drosophila".

Reviewer #3:

Remarks to the Author:

Combining optical tissue clearing, light-sheet ultramicroscopy and image analysis, authors present a new approach for whole-fly 3D imaging without physical sectioning. Authors claimed that their approach allowed 3D visualization of whole-body neuronal networks with isotropic single-cell resolution. The approach is interesting and potentially useful, even though none of the used technologies are new. However, the resolution of generated 3D images was not better than other available methods. Without seeing a major advance in methodology and/or significantly improved image quality, I am hesitating to support the publication at the current form. Followings are several specific issues need to be addressed.

Major concerns:

1. In Figures 2 and 6: The thickness of the light sheet ranges from 3 microns to 23 microns within 2mm length. The total length of a fly from head to tail is around 3 mm. At the fly orientation indicated in the Figure 6, how is it possible that the 23 micron-thickness light sheet produces single-neuron resolution at the edge?
2. In Figures 3-5: To show the advantage of the developed light-sheet ultramicroscopy, authors need

to compare their results and features of the technology to results generated by other methods for visualizing neuronal structures within an intact tissue. For example, using Xradia MicroXCT-200 X-ray imaging system, 3D internal structures in the whole-fly has been previously visualized at the resolution better than the current method (J. Comparative Neurology (2015) 523:1281–1295). Furthermore, home-made X-ray imaging system has reached single-neuron resolution (Journal of Physics: Conference Series 186 (2009) 012092 doi:10.1088/1742-6596/186/1/012092).

3. Page 11, lines 7-10: The statement should be rephrased. Previously, the head-array preparation has allowed confocal imaging of individual sensory neurons connecting between sensory organs and the brain (J. Neurogenetics (2015) 26, 157-168). Comparing with the head-array preparation for confocal imaging, what are the additional advantages of the developed light-sheet UM?

4. In Supplementary Figures 5 and 6: While GFP intensity is strong in the antenna sensory neurons (near the surface), fluorescence signal diminished along the axonal projections extending into the brain. Weak glomerulus signals in the antenna lobe within the brain do not support authors' claim that the established light-sheet UM allows visualization of long-range connections of sensory receptor neurons for the analysis of global neuronal circuits. It will help reader to evaluate the tool if authors provide high-resolution images of neuronal terminals in the antennal lobe and other sensory processing centers.

5. In Figure 4 (page 10, lines 16-19): When authors claimed that they can trace the projection of the R7 and R8 neurons from M3/M6 layers through the lamina into the retina, they need to show single neuron projection. The schematic drawing of single neurons in Figure 4g is inappropriate if it is from the prior knowledge rather than the current study.

6. Page 11, lines 10-11: The statement should be deleted. Confocal imaging of the whole fly is possible with newly developed 25X objective lens with high N.A. and long working distance (XLSLPLN25XGMP, NA = 1.0, WD = 8.0 mm, Olympus), if the whole fly is transparent.

7. In Figure 6: It is nice to see a 3D image with an isotropic resolution. Is this also working for high-resolution image of single neurons or at least cell bodies? What is the improvement in z resolution? Any limitations? What is the impact on the analysis of neuronal connectivity?

Overall, the title of "high-resolution" is misleading. Using CUBIC-X tissue clearing and light-sheet fluorescence microscopy, the whole mouse brain has been imaged at single cell resolution with clear and sharp images of individual neuronal fibers (see Nature Neuroscience (2018) doi:10.1038/s41593-018-0109-1). Though clearing fly cuticle is a different issue, the current study should at least show a similar degree of image quality and resolution.

Minor concerns:

8. Page 6, lines 19-22: "A substantial change in Solution-1 (see Online Methods) from the original CUBIC reagent-1 was the replacement of Quadrol® (N,N,N',N'-tetrakis(2-hydroxypropyl)-ethylenediamine) by THEED (2,2',2'',2'''- (Ethylenedinitrilo)-tetraethanol). But, in page 21, lines 9-11: "Solution-1 was prepared by mixing 8 wt% N,N,N',N'-Tetrakis(2-hydroxypropyl) ethylenediamine (Sigma-Aldrich, 87600-100ML), 5wt % Triton® X 100 (Roth, 3051.2) and 25 wt% Urea (Roth, X999.2)." Which is correct?

9. Page 7, lines 5-7: In Solution-2, meglumine diatrizoate has been previously used as a tissue clearing reagent. Citation of the original paper (J. Comparative Neurology (2001) 440:1–11) should be

given.

10. Page 7, Line 13: In Supplementary Figure 3, authors showed that neural structures on the antennal are better visualized with confocal imaging than light-sheet UM. On the other hand, authors believe that confocal imaging is not suitable for taking whole animal images (Page 3 line 14). Clearly, the statement is only true under certain conditions that need to be clarified.

11. In Figure 4d and Supplementary Figure 7a, the texture of these images appears different from others. The procedures of image processing should be specified.

12. Page 6, line 24: Is "Fig. b" a typo of "Fig. 1b"?

13. In Supplementary Figure 5, insets in "a" do not match with enlarged images in "c" and "e". All images should keep the same orientation.

Reviewer 1

Major comments:

- There is no mention or measurement of 3D volume changes after the clearing. Some clearing technique induce some significant and non-uniform volume changes in different part of the body. It would be important to measure this potential distortion from the clearing method.

- Although preservation of fluorescent signal after solution 2 appears to be good, there is no quantitative measurement of fluorescent signal loss before and after the clearing method. For example, author can dissect the fly into half, measure fluorescent signal in one side before clearing and the other side (anatomically matching) after clearing. This will help to understand quantitatively how much fluorescent signal loss is expected from the clearing procedure and to determine whether weakly expression of fluorescent signal will be still preserved

Thank you very much for these important points. We proceeded as suggested (e.g. dissected files into half) and we addressed both major comments in the results under “FlyClear leads to highly transparent samples with long-term signal preservation” and the new Supplementary Fig 1.

Comparison of the cleared and uncleared half of the same flies showed no change in volume. In the cleared (fly) half the signal was reduced compared to the non-treated (fly) half. However, in the cleared half the autofluorescence was reduced as well. As we used a low expression model and the autofluorescence was diminished we could clearly show that samples with a weak signal could be cleared and imaged.

Minor comments:

- At the end of the UM development description result section for Figure 2, it would be useful to include a sentence or two to explain what is benefit of having lower FWHM in the new configuration for the imaging quality.

Thank you for the remark. We added a sentence at the beginning of the UM development description to make the advantage of a lower FWHM (= thinner light sheet) clearer.

Miscellaneous

- typo in page 6 line 24, should be Fig 1b

Thank you for the remark. We corrected the mistake.

- page 13, line 4, typo

Thank you for the remark. We corrected the mistake.

- Figure 1B bottom row, no scale bar

Thank you for the remark. We added the scale bars.

Reviewer 2

1) It would be helpful if authors could include an image of the adult *Drosophila* before and after the clearing with FlyClear.

Thank you for this suggestion. We addressed this remark in Supplementary Fig.1

2) Authors should mention that the THEED used in FlyClear Solution 1 was originally described in the CUBIC paper.

We added a comment that THEED was already tested in the original CUBIC paper.

3) Figure 2 needs a better explanation for a general readership. It may be unclear to most readers whether z is for the detection lens or illumination lens. The schema might better include the detection system with axis information, in case some of the readers are not familiar with the ultramicroscopy. It is also important to mention the objective lenses and their imaging area (x-y) for the fluorescence detection, because this information directly relates to the z value in this figure.

Thank you very much for pointing out that our way of labelling parts of Figure 2 can be difficult to understand. We changed the labelling of “z” into “x” in Figure 2 b, c. Additionally, we added a complete new Supplementary Fig. 6 showing the imaging chamber, a scheme of the UM microscope with all the objectives used in this manuscript and the requested imaging area information.

4) Please provide the refractive index of FlyClear (solution 2 for mounting), as this is important information to correct for spherical aberrations.

The RI of *Solution-2* is 1.45. We provided this information in the results “FlyClear leads to highly transparent samples with long-term signal preservation” and in the Methods “FlyClear solution preparation” section.

5) It is not always apparent in the figures whether the image was taken with ultramicroscopy or confocal. The microscopy type should be labelled within each figure (with objective lens information). To improve the transparency of the paper, please include a table summarizing the imaging conditions (objective lens, ultramicroscopy/confocal, etc).

We marked the requested information in the figure legends in yellow. In addition, we provided a summary of the imaging conditions in Supplementary Table 2.

6) Related to 3), as many of the researchers in the field only have experiences with confocal microscopy, it would be helpful if authors could provide a side-by-side comparison (resolution, imaging speed, etc.) of confocal vs ultramicroscopy in the supplement.

We added the required data to the Supplementary Table 2.

7) In the introduction, authors should mention common mounting/clearing media that were previously used (e.g., Vectashield, FocusClear, SeeDB2, DPX) and their limitations.

We mentioned properties of a good RI and imaging medium in the results section “FlyClear leads to highly transparent samples with long-term signal preservation”. Further, we added a new Supplementary Fig. 2 showing the transparency achieved in *Drosophila* with different RI media.

8) Be consistent throughout the text on the use of technical terms, particularly for “ultramicroscopy” and “Drosophila”.

Thank you for the remark. We corrected the mistakes.

Reviewer 3

Major concerns:

1. In Figures 2 and 6: The thickness of the light sheet ranges from 3 microns to 23 microns within 2mm length. The total length of a fly from head to tail is around 3 mm. At the fly orientation indicated in the Figure 6, how is it possible that the 23 micron-thickness light sheet produces single-neuron resolution at the edge?

Thank you for the remark we will try to make the point clearer. First we illuminated the sample from two directions (Light sheet 1 and 2, figure below), second the light-sheet enters the flies lateral (not anterior posterior). Using this method for imaging, we have a light-sheet thickness of about 4-5 μ m at worst inside the sample.

2. In Figures 3-5: To show the advantage of the developed light-sheet ultramicroscopy, authors need to compare their results and features of the technology to results generated by other methods for visualizing neuronal structures within an intact tissue. For example, using Xradia MicroXCT-200 X-ray imaging system, 3D internal structures in the whole-fly has been previously visualized at the resolution better than the current method (J. Comparative Neurology (2015) 523:1281–1295). Furthermore, home-made X-ray imaging system has reached single-neuron resolution (Journal of Physics: Conference Series 186 (2009) 012092 doi:10.1088/1742-6596/186/1/012092).

Thank you for the comment, we added a statement in the introduction and included a citation.

Impressive images could be generated with MicrosXCT and other x-ray systems and it has become the most widely used technique in insect anatomy (Curr Opin Insect Sci. 2016 Dec;18:60-68. doi: 10.1016/j.cois.2016.09.004.). Additionally, it is very potent when it comes to large scale analysis of system architecture and volumetric data of model and non-model organisms.

However, specific labelling of cells is not possible in this kind of approach. We see a major advantage of light-sheet microscopy, where we can analyse and distinguish specifically labelled cell types on a large scale with high resolution. Further, we have to comment that x-ray imaging systems could reach-single-neuron resolution in mouse (Journal of Physics: Conference Series 186 (2009) 012092 doi:10.1088/1742-6596/186/1/012092), but mouse neurons are a magnitude larger (depicted neurons have cell bodies of 20µm) than *Drosophila* neurons. Furthermore, the resolution of x-ray imaging appears to be much lower when compared to light-sheet imaging of mouse neuronal structures with similar size (CLARITY, CUBIC, 3DISCO, uDISCO). Additionally, it has to be mentioned that the tissue surrounding the neurons is different. One of the main challenges in visualising deep structures with high resolution in *Drosophila* is imaging through the cuticle, which has very different light scattering properties compared to (mouse) brain tissue.

3. Page 11, lines 7-10: The statement should be rephrased. Previously, the head-array preparation has allowed confocal imaging of individual sensory neurons connecting between sensory organs and the brain (J. Neurogenetics (2015) 26, 157-168). Comparing with the head-array preparation for confocal imaging, what are the additional advantages of the developed light-sheet UM

Thank you for the remark. The head-array preparation could obtain remarkable images of the sensory neurons in fly heads. However this approach represents a very laborious and sophisticated way of dissecting the brain out of the flies, inheriting all the challenges common in dissection approaches e.g. information loss and possible tissue deformation. In addition, the connections in this publication could only be imaged in the cut plain, which can also be carried out without the described sample preparation. Further, in this publication the sample could just be recorded from coronal direction making it very difficult to image e.g. sensory neurons in the antenna projecting into the brain. In comparison UM microscopy has the major

advantage of imaging the sample (and the same area) from any direction, not being restricted exclusively to the head.

To address this concern we rephrased the statement in “**undissected** projection of sensory neurons”

4. In Supplementary Figures 5 and 6: While GFP intensity is strong in the antenna sensory neurons (near the surface), fluorescence signal diminished along the axonal projections extending into the brain. Weak glomerulus signals in the antenna lobe within the brain do not support authors’ claim that the established light-sheet UM allows visualization of long-range connections of sensory receptor neurons for the analysis of global neuronal circuits. It will help reader to evaluate the tool if authors provide high-resolution images of neuronal terminals in the antennal lobe and other sensory processing centers

Thank you for the remark. The image in old Supplementary Figure 5 (now Supplementary Fig. 8) just displays a clipping plane of a whole animal stack and it is correct that the structures appear weaker, but we were emphasising on different structures. To address this concern, we now included the whole stack image in the new Supplementary Figure 8 and you can find a cut out of a clipping plane in the Figure blow indicating a strong signal in the mentioned structures of the same stack. Additionally, we have to emphasize that the head of this sample was imaged with high resolution (25x 1.0 NA objective) shown in old Video 4 (now Video 5) and that the fluorescence signal in this Video appears strong in the mentioned structures.

Regarding the old Supplementary Figure 6 (now supplementary Figure 9), the remark is undoubtedly correct, however, this is due to the way the animal was recorded. In this Figure we were focusing on the distribution of mechanosensory neurons in legs and wings and we imaged with a low laser intensity and short exposition times to avoid over exposition. To address this concern we added an image of a fly with the same genotype (*Peb-Gal4 UAS-mCD8::GFP;;*) to the new Figure 5c and made an additional Video 7 where we screen through the whole head.

5. In Figure 4 (page 10, lines 16-19): When authors claimed that they can trace the projection of the R7 and R8 neurons from M3/M6 layers through the lamina into the retina, they need to show single neuron projection. The schematic drawing of single neurons in Figure 4g is inappropriate if it is from the prior knowledge rather than the current study.

We agree that the combined labelling of different photoreceptors in the medulla masks the layer-specific innervation. To illustrate the spatial resolution in the central optic neuropil we imaged a distinct population of medulla columnar neurons and could resolve the central synaptic layer from the peripheral cell body layer. We have included the images into Figure 4.

6. Page 11, lines 10-11: The statement should be deleted. Confocal imaging of the whole fly is possible with newly developed 25X objective lens with high N.A. and long working distance (XLSLPLN25XGMP, NA = 1.0, WD = 8.0 mm, Olympus), if the whole fly is transparent.

The statement was deleted.

7. In Figure 6: It is nice to see a 3D image with an isotropic resolution. Is this also working for high-resolution image of single neurons or at least cell bodies? What is the improvement in z resolution? Any limitations? What is the impact on the analysis of neuronal connectivity?

Thank you for the remark. Regarding the impact of the analysis of the neuronal connectivity we generated a completely new Fig 4a-d. We used the *nrg849/y ; UAS-mCD8::GFP ; ato-Gal4* to analyse interhemispheric circuit defects in *Neuroglian* mutants where the commissural connection of visual neurons was lost in comparison to the wild type.

To address the question of high resolution we imaged the *Neuroglian* mutants from orthogonal directions and we performed multiview combining. With this approach we could improve the z resolution by a factor of 3, but as mentioned in the manuscript, on the cost of the xy resolution. Nevertheless, the round shape of cell bodies can be clearly seen from all directions as well as the connections between the neurons (see Video 4).

Overall, the title of “high-resolution” is misleading. Using CUBIC-X tissue clearing and light-sheet fluorescence microscopy, the whole mouse brain has been imaged at single cell resolution with clear and sharp images of individual neuronal fibers (see Nature Neuroscience (2018) doi:10.1038/s41593-018-0109-1). Though clearing fly cuticle is a different issue, the current study should at least show a similar degree of image quality and resolution.

As mentioned in the response to concern 2, mouse neurons are a magnitude larger (factor 3-9) than *Drosophila* neurons. It is therefore not as challenging to achieve single cell resolution. In case of CUBIC-X, it belongs to the expansion techniques where swelling of the tissue is induced on purpose to increase the resolution. To give an example of size difference, a neuron visualized in the CUBIC-X paper has a cell body size of about 35µm. In comparison, an average neuronal cell body in *Drosophila* has a size of 2-6µm (DOI: 10.1523/JNEUROSCI.3348-09.2009). Since we observed no change in size and tissue morphology of *Drosophila* (Supplementary Fig. 1) this means that the neurons described in the CUBIC-X paper are about a factor 6 - 12.5 larger.

In addition to the size of *Drosophila* neurons, the pigment and the cuticle are the exact reason why deep tissue imaging in this model organism is so challenging.

In this revised manuscript we could clearly image cell bodies and their projections in deep *Drosophila* tissue of undissected samples. Therefore, we believe that the term “high-resolution” in the title is justified.

Minor concerns:

8. Page 6, lines 19-22: “A substantial change in Solution-1 (see Online Methods) from the original CUBIC reagent-1 was the replacement of Quadrol® (N,N,N',N'-tetrakis(2-hydroxypropyl)-ethylenediamine) by THEED (2,2',2'',2'''- (Ethylenedinitrilo)-tetraethanol). But, in page 21, lines 9-11: “Solution-1 was prepared by mixing 8 wt% N,N,N',N'-Tetrakis(2-hydroxypropyl) ethylenediamine (Sigma-Aldrich, 87600-100ML), 5wt % Triton® X 100 (Roth, 3051.2) and 25 wt% Urea (Roth, X999.2).” Which is correct?

Thank you very much for the remark. We corrected the mistake.

9. Page 7, lines 5-7: In Solution-2, meglumine diatrizoate has been previously used as a tissue clearing reagent. Citation of the original paper (J. Comparative Neurology (2001) 440:1–11) should be given.

In the publication (J. Comparative Neurology (2001) 440:1–11) FocusClear™ was used as a clearing agent which consists of at least four different components one of which is meglumine diatrizoate (not specified in this publication). Additionally, *Solution-2* performed better than

FocusClear in *Drosophila* tissue (Figure 1b and Supplementary Fig.2). Nevertheless we added a reference for the mentioned paper.

10. Page 7, Line 13: In Supplementary Figure 3, authors showed that neural structures on the antennal are better visualized with confocal imaging than light-sheet UM. On the other hand, authors believe that confocal imaging is not suitable for taking whole animal images (Page 3 line 14). Clearly, the statement is only true under certain conditions that need to be clarified.

The aim of the old Supplementary Figure 3 (new Supplementary Fig. 5) was to show, that the clearing approach is compatible with a second fluorophore (mCherry). The differences in the image resolution are because of the different objectives used (10x 0.3NA and a 0.5x demagnification for UM microscopy and 20x 0.7NA for confocal microscopy). Since the 20x objective has a working distance of 260µm and the fly has an anterior posterior size of 3000µm and a dorso ventral size of 1000-2000µm (depending if the legs are included in the imaging) whole fly imaging with such an objective is not feasible. As mentioned before, in comment 6, long working distance objectives could image a whole fly sample. Nevertheless, the long acquisition time, large data volume and the photo bleaching still represent main challenges of confocal microscopy for large sample imaging (Cell. 2015 Jul 16;162(2):246-257. doi: 10.1016/j.cell.2015.06.067). Therefore, we believe that UM is more suitable for whole system approaches.

11. In Figure 4d and Supplementary Figure 7a, the texture of these images appears different from others. The procedures of image processing should be specified.

We performed a stronger histogram equilibration to achieve a more even distribution of intensities, which is a standard procedure in image processing.

12. Page 6, line 24: Is “Fig. b” a typo of “Fig. 1b”?

Thank you for the remark. We corrected the mistake.

13. In Supplementary Figure 5, insets in “a” do not match with enlarged images in “c” and “e”. All images should keep the same orientation.

Dashed rectangles were removed from old Figure 5a (new Supplementary Fig. 8) to keep the current orientation of the images.

Reviewers' Comments:

Reviewer #1:

Remarks to the Author:

I would like to thank authors to perform both volume and fluorescent signal measurement before and after the clearing method. I'm satisfied with the revised manuscript and would like to support the acceptance of the manuscript.

Reviewer #2:

Remarks to the Author:

Unfortunately, authors have not adequately addressed some of my previous comments.

2) Supplementary Fig. 6 is a welcome addition, but Figure 2 has not been improved at all. Please include the xyz axes in panel a. Also indicate the sample position in a. Where is the $x = 0$ position?

7) Please respond to my previous comment 7. Please explain what kinds of mounting media have been previously used, and what have been limitations of these media.

8) Please seriously respond to my previous comment 8. Ultramicroscopy is sometimes UM or ultramicroscopy (UM). Please describe as "ultramicroscopy (UM)" for the first use, and just use "UM" thereafter; alternatively, authors could avoid use of "UM". Similarly, *Drosophila* is sometimes non-italic, *Drosophila melanogaster*, or *D. melanogaster*. FlyClear is italic in somewhere, but non-italic in elsewhere. Be consistent in terminology to improve the readability.

In addition, authors have not yet addressed to a major comment by Reviewer 1, which I agree to be important. Authors argue that FlyClear is improved over previous methods, including CUBIC. As a methods paper, authors need to demonstrate the improvement based on a quantitative metric. Supplementary Fig. 1C is only showing the reduction of autofluorescence, not the preservation of GFP. Authors could use the dissected brain, for example, to show the stability of GFP, if the autofluorescence in the cuticle is problematic.

Reviewer #3:

Remarks to the Author:

The study combined several technological developments including optical tissue clearing, light-sheet ultramicroscopy and image analysis for whole-fly 3D imaging without physical sectioning. While authors have addressed some of my previous concerns, one major concern remains. While removal of cuticle pigments is a valuable technique for deep tissue optical imaging, I feel strongly that the "high-resolution" title is an overstatement of the advance because X-ray tomography has achieved better resolution without removing cuticle pigments.

Authors argued that specific labeling of neurons is not possible for X-ray imaging. In fact, X-ray tomography has been used to three-dimensionally visualize *Drosophila* brain neurons genetically labeled with minSOG (Ng et al. 2016). Authors also argued that mouse neurons are much larger than fly neurons and thus it is challenging to reach single neuron resolution in flies. In fact, X-ray tomography has been established for high-speed three-dimensional reconstruction of *Drosophila* brain neurons at subcellular resolution without removing cuticle pigments (Hwu et al. 2017). Authors should properly discuss the significance of their results by comparison to existing techniques and tone down

their claims.

References:

Julian Ng, Alyssa Browning, Lorenz Lechner, Masako Terada, Gillian Howard, and Gregory S. X. E. Jefferis (2016) Genetically targeted 3D visualization of Drosophila neurons under electron microscopy and X-ray microscopy using miniSOG. *Sci Rep.* 6: 38863.

Yeukuang Hwu, Giorgio Margaritondo and Ann-Shyn Chiang (2017) Why use synchrotron x-ray tomography for multi-scale connectome mapping? *BMC Biology* 15:122.

Reviewer #2

Remarks to the Author:

2) Supplementary Fig. 6 is a welcome addition, but Figure 2 has not been improved at all. Please include the xyz axes in panel a. Also indicate the sample position in a. Where is the x = 0 position?

We changed panel a) of Figure 2 completely and indicated the axis of the light sheet in panel a), b) and c) to make the whole figure more understandable.

7) Please respond to my previous comment 7. Please explain what kinds of mounting media have been previously used, and what have been limitations of these media.

We added a new part in the introduction, mentioning mounting media with cross references to literature describing their limitations. A lot of the mentioned mounting media were already discussed in the SeeDB2 publication.

8) Please seriously respond to my previous comment 8. Ultramicroscopy is sometimes UM or ultramicroscopy (UM). Please describe as “ultramicroscopy (UM)” for the first use, and just use “UM” thereafter; alternatively, authors could avoid use of “UM”. Similarly, *Drosophila* is sometimes non-italic, *Drosophila melanogaster*, or *D. melanogaster*. FlyClear is italic in somewhere, but non-italic in elsewhere. Be consistent in terminology to improve the readability.

We are sorry for missing the opportunity to perform these changes in the last revision.

-We changed all *Drosophila melanogaster*, *D. melanogaster* and *Drosophila* to italic

-We avoided UM completely and used the full term

-We changed FlyClear to non-italic

In addition, authors have not yet addressed to a major comment by Reviewer 1, which I agree to be important. Authors argue that FlyClear is improved over previous methods, including CUBIC. As a methods paper, authors need to demonstrate the improvement based on a quantitative metric. Supplementary Fig. 1C is only showing the reduction of autofluorescence, not the preservation of GFP. Authors could use the dissected brain, for example, to show the stability of GFP, if the autofluorescence in the cuticle is problematic.

Regarding this comment we believe there might be a confusion. Supplementary Figure 1b shows an example of an uncleared and a cleared antenna from the same fly. There, the GFP signal is still visible in the cleared antenna, although a significant fluorescent loss is apparent (both antennae were imaged using the same laser intensity). The quantification of this GFP loss can be seen in Supplementary Figure 1c, where we compared the signal in the antennae of three uncleared fly

half's with their corresponding cleared half's (all imaged with same laser intensity). Again we measured a statistically significant loss of GFP signal.

To understand the design of the experiment and to understand why we favoured the antenna in contrast to dissected brains or any other organs some explanation is necessary.

To demonstrate that we can still image the neurons although there is an initial signal loss, we used a model with very weak expression (in our case 50 neurons in each antenna). This should convince the reader that even processing of samples with low level of fluorescence expression is possible. Further, we wanted to quantify the signal through the cuticle. For this we used the antenna, since we could easily image the GFP signal with confocal microscopy without initial tissue clearing. Finally, we are limited by the working distance of the objectives used with our confocal microscope (the long working distance objectives we used for light sheet imaging are from Olympus and our confocal is from Leica).

The major improvement of our approach in comparison to the original CUBIC approach is the ability to depigment the eyes of *Drosophila*, the higher transparency of the cleared sample (Supplementary Figure 2) and the long signal preservation after clearing.

Reviewer #3

Remarks to the Author:

The study combined several technological developments including optical tissue clearing, light-sheet ultramicroscopy and image analysis for whole-fly 3D imaging without physical sectioning. While authors have addressed some of my previous concerns, one major concern remains. While removal of cuticle pigments is a valuable technique for deep tissue optical imaging, I feel strongly that the "high-resolution" title is an overstatement of the advance because X-ray tomography has achieved better resolution without removing cuticle pigments.

Authors argued that specific labeling of neurons is not possible for X-ray imaging. In fact, X-ray tomography has been used to three-dimensionally visualize *Drosophila* brain neurons genetically labeled with minSOG (Ng et al. 2016). Authors also argued that mouse neurons are much larger than fly neurons and thus it is challenging to reach single neuron resolution in flies. In fact, X-ray tomography has been established for high-speed three-dimensional reconstruction of *Drosophila* brain neurons at subcellular resolution without removing cuticle pigments (Hwu et al. 2017). Authors should properly discuss the significance of their results by comparison to existing techniques and tone down their claims.

-Regarding the title "high resolution" we believe that this phrase is not overstated. The critics would be well addressed in case we had claimed subcellular resolution e.g. boutons. We showed distinguishable cell bodies in dense neuronal clusters and the fine branching of their projections. Therefore, we believe the term high resolution is appropriate, despite other approaches might display even higher level of detail.

-Thank you for raising our attention to the point that genetically labelled cells can be marked specifically with high x-ray contrast materials. However, such labelling has major limitations, which makes whole system approaches or even thick tissue (>200µm) labelling at the current state nearly impossible. Firstly, such labelling approaches either have limited targeting capabilities (e.g. HRP)¹⁻³, show the necessity for cell-toxic conditions for probe introduction (e.g. ReASH)⁴, or lead to neurotoxicity over time (e.g. miniSOG)⁵. Secondly, in case of miniSOG, the reaction producing the contrast material for x-ray is induced by fluorescent light. If the tissue between the genetically marked target area and the light source is not transparent, it is very difficult to start the labelling reaction in deep tissue, since the light is scattered. Ng et al. could label with the miniSOG approach tissue of about 100µm after brain dissection⁵. This would be even less, if a pigmented cuticle scatters the light beforehand. Thirdly, since the light induced generation of the contrast materials has to be stopped at a certain point and various parameters such as probe expression level, localised targeting, labelling density, fixation conditions, DAB photo-oxidation and microscope setting's play a role for the optimal reaction duration, it is very difficult to find the delicate balance between under- and overlabeling of the sample⁵.

-Regarding the establishment of x-ray tomography for high-speed three-dimensional reconstruction of *Drosophila* brain neurons at subcellular resolution without removing cuticle pigments (Hwu et al. 2017)⁶; we agree that impressive results can be achieved with such an approach. Nevertheless, there are certain limitations to this matter. As specifically discussed above, dissection-free whole system labelling is not possible at the current state. Golgi stained neurons can certainly be visualised with high resolution, however, this labelling occurs in a random fashion and the process of the uptake of the heavy metal ions inside the cell body of the neurons is not completely understood yet⁶. Further, this approach is not without distortion. When looking at Fig. 2a in Hwu et al. it becomes apparent that the abdomen and thorax of the depicted fly are deformed⁶. Finally, a synchrotron facility is very cost intensive and not accessible to the broad majority of scientists.

We understand the importance of x-ray imaging approaches. The strength of our approach is the systemic- or high resolution deep tissue imaging of specific labelled structures. We believe that our claims are not out of proportion.

We cited the requested references in the introduction and added a part dealing with x-ray imaging to the discussion.

References

1. Watts, R.J., Schuldiner, O., Perrino, J., Larsen, C. & Luo, L. Glia engulf degenerating axons during developmental axon pruning. *Current biology : CB* **14**, 678-684 (2004).
2. Li, J., Wang, Y., Chiu, S.L. & Cline, H.T. Membrane targeted horseradish peroxidase as a marker for correlative fluorescence and electron microscopy studies. *Frontiers in neural circuits* **4**, 6 (2010).
3. Atasoy, D. et al. A genetically specified connectomics approach applied to long-range feeding regulatory circuits. *Nature neuroscience* **17**, 1830-1839 (2014).
4. Boassa, D. et al. Mapping the subcellular distribution of alpha-synuclein in neurons using genetically encoded probes for correlated light and electron microscopy: implications for Parkinson's disease pathogenesis. *The Journal of neuroscience : the official journal of the Society for Neuroscience* **33**, 2605-2615 (2013).
5. Ng, J. et al. Genetically targeted 3D visualisation of Drosophila neurons under Electron Microscopy and X-Ray Microscopy using miniSOG. *Sci Rep* **6**, 38863 (2016).
6. Hwu, Y., Margaritondo, G. & Chiang, A.S. Q&A: Why use synchrotron x-ray tomography for multi-scale connectome mapping? *BMC biology* **15**, 122 (2017).